# EFFICIENT OFF-POLICY META-REINFORCEMENT LEARNING VIA PROBABILISTIC CONTEXT VARIABLES

**Kate Rakelly\*, Aurick Zhou\*, Deirdre Quillen, Chelsea Finn, Sergey Levine** Department of Electrical Engineering an
UC Berkeley
Berkeley CA, 94709
`{rakelly, azhou, dquillen, cbfinn, svlevine}@eecs.berkeley.edu`

## 1 INTRODUCTION

Learning large repertoires of behaviors with conventional RL methods quickly becomes prohibitive as learning each task often requires millions of interactions with the environment. Fortunately, many of the problems we would like our autonomous agents to solve share common structure. For example screwing a cap on a bottle and turning a doorknob both involve grasping an object in the hand and rotating the wrist. Exploiting this structure to learn new tasks more quickly remains an open and pressing topic.

While meta-learned policies adapt to new tasks with only a few trials, during training they require massive amounts of data drawn from a large set of distinct tasks, exacerbating the problem of sample efficiency that plagues RL algorithms. Most current meta-RL methods require on-policy data during both meta-training and adaptation Finn et al. (2017); Wang et al. (2016); Duan et al. (2016); Mishra et al. (2018); Rothfuss et al. (2018); Houthooft et al. (2018), rendering them exceedingly inefficient during meta-training. However, making use of off-policy data for meta-RL poses new challenges. Meta-learning typically operates on the principle that meta-training time should match meta-test time. This makes it inherently difficult to meta-train a policy to adapt from off-policy data, which is systematically different from the data the policy would see when it explores (on-policy) in a new task at meta-test time.

To achieve both adaptation and meta-training data efficiency, our approach integrates online inference of probabilistic context variables with existing off-policy RL algorithms. During meta-training, we learn a probabilistic encoder that accumulates the necessary statistics from past experience that enable the policy to perform the task. At meta-test time, our method adapts quickly by sampling context variables ("task hypotheses"), acting according to that task, and then updating its belief about the task by updating the posterior over the context variables. Our approach integrates easily with existing off-policy RL algorithms, enabling good sample efficiency during meta-training.

The primary contribution of our work is an off-policy meta-RL algorithm Probabilistic Embeddings for Actor-critic RL (PEARL) that achieves excellent sample efficiency during meta-training, enables fast adaptation by accumulating experience online, and performs structured exploration by reasoning about uncertainty over tasks. We demonstrate 20-100X improvement in meta-training sample efficiency on six continuous control meta-learning environments, and demonstrate how our model structured exploration to adapt rapidly to new tasks with sparse rewards.

## 2 RELATED WORK

Our work builds on the meta-learning framework Schmidhuber (1987); Bengio et al. (1990); Thrun & Pratt (1998) in the context of reinforcement learning. Recurrent Duan et al. (2016); Wang et al. (2016) and recursive Mishra et al. (2018) meta-RL methods adapt to new tasks by aggregating experience into a latent representation on which the policy is conditioned. We model latent task variables as probabilistic and use a simpler aggregation function inspired by Snell et al. (2017). Prior work has explored training recurrent Q-functions with off-policy Q-learning methods Heess et al. (2015); Hausknecht & Stone (2015). We find the straightforward application of these methods to meta-RL difficult, and explore how to effectively make use of off-policy data during meta-training. Gradient-based meta-RL methods focus on on-policy learning, using policy gradients Finn et al. (2017); Stadie et al. (2018); Rothfuss et al. (2018); Xu et al. (2018a), meta-learned loss functions

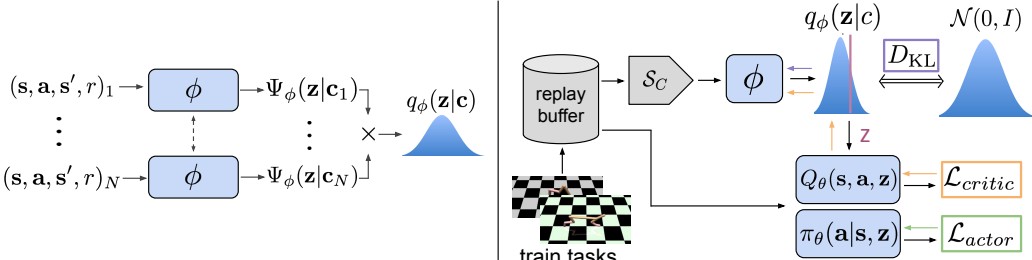

Figure 1: (left) The inference network predicts the posterior over the latent context variables $q_\phi(\mathbf{z}|\mathbf{c})$ as a permutation-invariant function of prior experience. Samples from this posterior condition the policy. (right) The actor and critic are meta-learned jointly with the inference network, which is optimized with gradients from the critic as well as from an information bottleneck on $\mathcal{Z}$. De-coupling the data sampling strategies for context and RL batches is critical for off-policy learning.

Sung et al. (2017); Houthooft et al. (2018), or hyperparameters Xu et al. (2018b). We instead focus on meta-learning from off-policy data, which is non-trivial to do with these prior methods.

Prior work has applied probabilistic models to meta-learning. For supervised learning, Rusu et al. (2019); Gordon et al. (2019); Finn et al. (2018) adapt model predictions using probabilistic latent task variables inferred via amortized approximate inference. In RL, Hausman et al. (2018) also conditions the policy on inferred task variables, but the aim is to compose skills via the learned embedding space, while we focus on adapting to new tasks. While we infer task variables and explore via posterior sampling, Gupta et al. (2018) adapts via gradient descent and explores via sampling from the prior.

## 3 METHOD

We assume a distribution of tasks $p(\mathcal{T})$, where each task is a Markov decision process (MDP). Formally, a task $\mathcal{T} = \{p(\mathbf{s}_0), p(\mathbf{s}_{t+1}|\mathbf{s}_t, \mathbf{a}_t), r(\mathbf{s}_t, \mathbf{a}_t)\}$ consists of an initial state distribution $p(\mathbf{s}_0)$, transition distribution $p(\mathbf{s}_{t+1}|\mathbf{s}_t, \mathbf{a}_t)$, reward function $r(\mathbf{s}_t, \mathbf{a}_t)$. We assume that the transition and reward functions are unknown, but can be sampled by taking actions in the environment. Given a set of training tasks sampled from $p(\mathcal{T})$, the meta-training process learns a policy that adapts to the task at hand by conditioning on the history of past transitions, which we refer to as *context* $C$. Let $\mathbf{c}_n^l = (\mathbf{s}_n, \mathbf{a}_n, r_n, \mathbf{s}_n')$ be one transition in the task $l$ so that $\mathbf{c}_{1:N}^l$ comprises the experience collected so far. At test-time, the policy must adapt to a new set of tasks from $p(\mathcal{T})$.

### 3.1 PROBABILISTIC LATENT CONTEXT

We capture knowledge about how the current task should be performed in a latent probabilistic context variable $Z$, on which we condition the policy as $\pi_\theta(\mathbf{a}|\mathbf{s}, \mathbf{z})$. Meta-training consists of leveraging data from a variety of training tasks to learn to infer $Z$ from a recent history of experience in the new task, as well as optimizing the policy to solve the task given samples from the posterior over $Z$.

To enable adaptation, the latent context $Z$ must encode salient information about the task. We adopt an amortized variational inference approach Kingma & Welling (2014); Rezende et al. (2014); Alemi et al. (2016) to learn to infer $Z$. We train an *inference network* $q_\phi(\mathbf{z}|\mathbf{c})$ that estimates the posterior $p(\mathbf{z}|\mathbf{c})$. While there are several choices for the objective to optimize $q_\phi(\mathbf{z}|\mathbf{c})$ including learning predictive models of rewards and dynamics or maximizing returns through the policy, we choose to optimize it to predict the task state-action value function. Modeling the objective as a pseudo-likelihood, the resulting variational lower bound training objective is:

$$\mathbb{E}_{\mathcal{T}}[\mathbb{E}_{\mathbf{z} \sim q(\mathbf{z}|\mathbf{c}^{\mathcal{T}})}[R(\mathcal{T}, \mathbf{z}) + \beta D_{\text{KL}}(q(\mathbf{z}|\mathbf{c}^{\mathcal{T}})||p(\mathbf{z}))]] \tag{1}$$

where $p(\cdot)$ is a unit Gaussian prior over $Z$ and $R(\mathcal{T}, \mathbf{z})$ is the Bellman error for a state-action value function conditioned on $\mathbf{z}$. While the parameters of $q_\phi$ are optimized during meta-training, at meta-test time the latent context for a new task is simply inferred from gathered experience.

The inference network $q_\phi(\mathbf{z}|\mathbf{c})$ should be expressive enough to capture minimal sufficient statistics of task-relevant information, without modeling irrelevant dependencies. We note that an encoding

of a fully observed MDP should be permutation invariant: if we would like to infer what the task is, identify the MDP model, or train a value function, it is enough to have access to a collection of transitions $\{\mathbf{s}_i, \mathbf{a}_i, \mathbf{s}'_i, r_i\}$, without regard for the order in which these transitions were observed. We therefore choose a permutation-invariant representation $q_\phi(\mathbf{z}|\mathbf{c}_{1:N})$ factorized as

$$q(\mathbf{z}|\mathbf{c}_{1:N}) \propto \Pi_{n=1}^N \Psi(\mathbf{z}|\mathbf{c}_n) \tag{2}$$

To keep the method tractable, we use Gaussian factors $\Psi(\mathbf{z}|\mathbf{c}_n) = \mathcal{N}(f^\mu(\mathbf{c}_n), f^\sigma(\mathbf{c}_n))$, which result in a Gaussian posterior, see Figure 1 (left).

For fast adaptation at meta-test time, it is critical for the agent to be able to explore and determine the task efficiently. In prior work, posterior sampling for exploration Strens (2000); Osband et al. (2013) models a distribution over MDPs and executes the optimal policy for an MDP sampled from the posterior for the duration of an episode. Acting optimally according to a random MDP allows for temporally extended exploration, meaning that the agent can act to test hypotheses even when the results of actions are not immediately informative of the task. PEARL meta-learns a prior over $Z$ that captures the distribution over tasks. Sampling $\mathbf{z}$'s (initially from the prior and then the updated posterior) and holding them constant across an episode results in temporally extended exploration strategies which become closer to the optimal behavior for the task as the belief narrows.

## 3.2 OFF-POLICY META-REINFORCEMENT LEARNING

A primary goal of our work is to enable efficient off-policy meta-reinforcement learning. However, designing off-policy meta-RL algorithms is non-trivial partly because modern meta-learning is predicated on the assumption that the distribution of data used for adaptation will match across meta-training and meta-test. In RL, this implies that since at meta-test time on-policy data will be used to adapt, on-policy data should be used during meta-training as well. Furthermore, meta-RL requires the policy to reason about *distributions* to learn effective stochastic exploration strategies. This problem inherently cannot be solved by off-policy RL methods that minimize temporal-difference error, as they do not have the ability to directly optimize for distributions of states visited. In contrast, policy gradient methods have direct control over the actions taken by the policy.

Our main insight in designing an off-policy meta-RL method with the probabilistic model in Section 3.1 is that the data used to train the probabilistic encoder need not be the same as the data used to train the policy. The policy can treat the context $\mathbf{z}$ as part of the state in an off-policy RL loop, while the stochasticity of the exploration process is provided by the uncertainty in the encoder $q(\mathbf{z}|\mathbf{c})$. The actor and critic are always trained with off-policy data sampled from the entire replay buffer $\mathcal{B}$. We define a sampler $\mathcal{S}_\mathbf{c}$ to sample context batches for training the encoder - we find the sampling from a pool of recently collected data works best. We summarize our training procedure in Figure 1 (right).

We build our algorithm on top of the soft actor-critic algorithm (SAC) Haarnoja et al. (2018), an off-policy actor-critic method based on the maximum entropy RL objective which augments the traditional sum of discounted returns with the entropy of the policy.

We optimize the parameters of the inference network $q(\mathbf{z}|\mathbf{c})$ jointly with the parameters of the actor $\pi_\theta(\mathbf{a}|\mathbf{s}, \mathbf{z})$ and critic $Q_\theta(\mathbf{s}, \mathbf{a}, \mathbf{z})$, using the reparameterization trick to compute gradients for parameters of $q_\phi(\mathbf{z}|\mathbf{c})$ through sampled $\mathbf{z}$'s. We train the inference network using gradients from the Bellman update for the critic, given by the following loss function

$$\mathcal{L}_{critic} = \mathbb{E}_{\substack{(\mathbf{s},\mathbf{a},r,\mathbf{s}') \sim \mathcal{B} \\ \mathbf{z} \sim q_\phi(\mathbf{z}|\mathbf{c})}} (Q_\theta(\mathbf{s},\mathbf{a},\mathbf{z}) - (r + \bar{V}(\mathbf{s}', \bar{\mathbf{z}})))^2 \tag{3}$$

where $\bar{V}$ is a target network and $\bar{\mathbf{z}}$ indicates that gradients are not being computed through it.

## 4 EXPERIMENTS

**Sample Efficiency and Performance** We evaluate PEARL on six continuous control environments simulated via MuJoCo Todorov et al. (2012). These locomotion task distributions require adaptation across dynamics (Walker-2D-Params) or across reward functions (the rest of the domains), and were introduced by Finn et al. (2017) and Rothfuss et al. (2018). We compare to existing policy gradient meta-RL methods ProMP Rothfuss et al. (2018), MAML-TRPO Finn et al. (2017), and RL$^2$ Duan

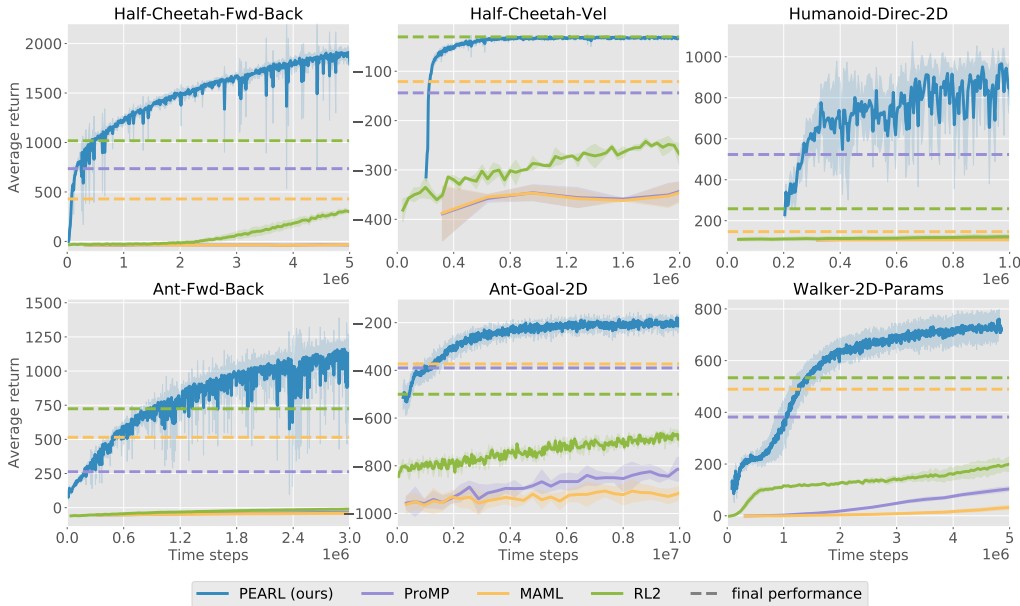

Figure 2: Test-task performance vs. samples collected during *meta-training* on continuous control domains. Dashed lines correspond to the maximum return achieved by each baseline after 1e8 steps. By leveraging off-policy data during meta-training, PEARL is $20-100$x more sample efficient than the baselines, and achieves state-of-the-art final performance.

et al. (2016) with PPO Schulman et al. (2017). We attempted to adapt recurrent DDPG Heess et al. (2015) to our setting, however we were unable to optimize it.

To evaluate on the meta-testing tasks, we perform online adaptation at the trajectory level, where the first trajectory is collected with context variable $z$ sampled from the prior $p(z)$ and subsequent trajectories are collected with $z \sim q(z|c)$. Here we report performance after two trajectories.

Our approach uses 20-100x fewer samples during meta-training than previous policy gradient approaches while often also improving final asymptotic performance, Figure 2.

**Posterior Sampling For Exploration** Posterior sampling in our model enables effective exploration strategies in sparse reward MDPs. We demonstrate this behavior with a 2-D navigation task in which a point robot must navigate to different locations on a semi-circle. A shaped reward is given only when the agent is within a certain radius of the goal (we experiment with radius 0.2 and 0.8). We sample training and testing sets of tasks, each consisting of 100 randomly sampled goals. To mitigate the difficulty of meta-training with sparse rewards, we assume access to the dense reward during meta-training, as in Gupta et al. (2018), but this burden could also be mitigated with task-agnostic exploration strategies.

In this setting, we compare to MAESN (Gupta et al. (2018)) and demonstrate we are able to adapt to the new sparse goal in fewer trajectories, while also requiring far fewer samples for meta-training to solve the task, Figure 3.

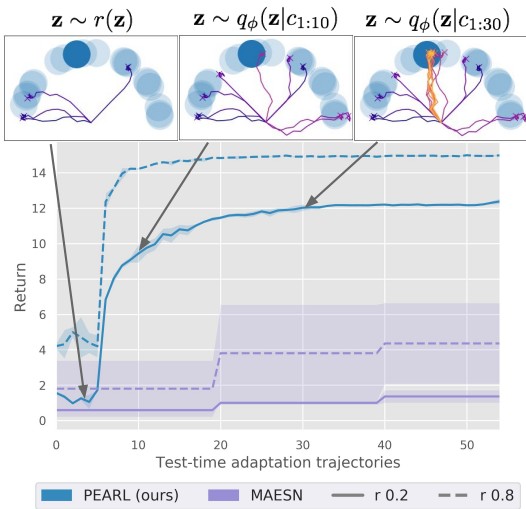

Figure 3: Sparse 2D navigation test-time adaptation. PEARL is able to start adapting to the task after collecting on average only 5 trajectories. We compare to MAESN ( Gupta et al. (2018)).

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
