# OpenReview forum: "Efficient Off-Policy Meta-Reinforcement Learning via Probabilistic Context Variables"
_ICLR.cc/2019/Workshop/LLD — LLD 2019_

### Official Review · AnonReviewer2 · 2019-04-09
**Important Meta-RL work but unsure if it fits the theme of the workshop**

**Rating:** 4
**Confidence:** 3

**Review:**

Note: While I really like the paper, I am not sure if it aligns with the workshop theme. The paper itself mentions "Under review at the Workshop on “Structure & Priors in Reinforcement Learning” at ICLR 2019" which I feel might be a better venue for this paper. I will let the organizers take a call and provide an objective assessment of the paper below without the workshop theme in consideration.

This paper proposes using off-policy RL during the meta-training time to greatly improve sample efficiency of Meta-RL methods.

Current meta-RL methods (MAML/ProMP) are terribly sample inefficient since the adaptation phase during training is trained using vanilla policy gradients with a linear feature baseline (which is known to be very sample inefficient). A natural next step would have been to use a learned state dependent baseline (analogous to actor-critic methods) by meta-learning a critic. The authors take a step further and use an off-policy RL method (SAC) which has shown to be better performing than on-policy actor critic methods. The policy is also conditioned on a task embedding, enabling better adaptation.

If the authors could clarify the following points, I think it would make the paper much better:
* It's not clear if this method could be directly extendible to partially-observed settings since the factorization of the inference network would no longer hold. If being restricted to fully observed domains is a limitation, it should be noted clearly in the paper.
* Ablations dissecting which of the two main components of the paper (inference network / using SAC) helped the performance most would be helpful, right now the relative importance of each component is unclear. Relatedly, the authors could also include a baseline a MAML/ProMP baseline where the critic is meta-learned.
* The experimental details in this version of the paper are severely limited. I realize the page length constraints but the authors could put details in the appendix.

---

### Official Review · AnonReviewer1 · 2019-04-12
**Interesting work, re-wraped from the original version a bit too fast**

**Rating:** 4
**Confidence:** 2

**Review:**

This paper tackles the problem of meta-learning, more precisely the possibility of considering off-policies. For this, a first network estimates, from the history on the current task (the "context"), which kind of task it is (represented by a variable 'z'), and a second network (in a standard actor-critic approach) learns the policy for the current task, thanks to a conditioning on this 'z' (which allows to perform transfer learning between similar tasks within the same network).

Experiments are performed on a standard problem (MuJoCo) and significant gains are observed (from 20 to 100 fewer examples needed for training).
Which justifies the submission to this workshop on Limited Labeled Data, even though the primary objective of the article is not really to deal with a given dataset of limited data, but rather to target sampling data efficiency, as the article is about RL tasks.

Overall, the paper is very good, with very interesting ideas; however it is clearly a very hastily made short version of a longer paper: abstract and conclusion were removed (are fully missing), text density maybe too high for a non-specialist.

Therefore I hesitate between "very good" and "borderline".

---

### Decision · Program_Chairs · 2019-04-16
**Acceptance Decision**

Accept